# Non-Mouse Models of Atherosclerosis: Approaches to Exploring the Translational Potential of New Therapies

**DOI:** 10.3390/ijms232112964

**Published:** 2022-10-26

**Authors:** Danielle Kamato, Iqra Ilyas, Suowen Xu, Peter J. Little

**Affiliations:** 1Discovery Biology, Griffith Institute for Drug Discovery, School of Environment and Science, Griffith University, Brisbane, QLD 4111, Australia; 2Pharmacy Australia Centre of Excellence, School of Pharmacy, University of Queensland, Woolloongabba, QLD 4102, Australia; 3Laboratory of Metabolics and Cardiovascular Diseases, University of Science and Technology of China, Hefei 230027, China; 4Biomedical Sciences and Health Laboratory of Anhui Province, University of Science and Technology of China, Hefei 230027, China; 5Department of Endocrinology, Institute of Endocrine and Metabolic Diseases, The First Affiliated Hospital of USTC, Division of Life Sciences and Medicine, Clinical Research Hospital of Chinese Academy of Sciences (Hefei), University of Science and Technology of China, Hefei 230001, China; 6Sunshine Coast Health Institute and School of Health and Behavioural Sciences, University of the Sunshine Coast, Birtinya, QLD 4575, Australia

**Keywords:** key atherosclerosis, animal models, therapeutics, disease mechanism, pathology

## Abstract

Cardiovascular disease is the largest single cause of disease-related mortality worldwide and the major underlying pathology is atherosclerosis. Atherosclerosis develops as a complex process of vascular lipid deposition and retention by modified proteoglycans, endothelial dysfunction and unresolved chronic inflammation. There are a multitude of current therapeutic agents, most based on lowering plasma lipid levels, but, overall, they have a lower than optimum level of efficacy and many deaths continue to arise from cardiovascular disease world-wide. To identify and evaluate potential novel cardiovascular drugs, suitable animal models that reproduce human atherosclerosis with a high degree of fidelity are required as essential pre-clinical research tools. Commonly used animal models of atherosclerosis include mice (ApoE^−/−^, LDLR^−/−^ mice and others), rabbits (WHHL rabbits and others), rats, pigs, hamster, zebrafish and non-human primates. Models based on various wild-type and genetically modified mice have been extensively reviewed but mice may not always be appropriate. Thus, here, we provide an overview of the advantages and shortcomings of various non-mouse animal models of atherosclerotic plaque formation, and plaque rupture, as well as commonly used interventional strategies. Taken together, the combinatorial selection of suitable animal models readily facilitates reproducible and rigorous translational research in discovering and validating novel anti-atherosclerotic drugs.

## 1. Introduction

Cardiovascular disease (CVD) is the largest single cause of morbidity worldwide and its major underlying pathology is atherosclerosis [1,2,3,4]. Atherosclerosis is the multifactorial chronic inflammatory disease of certain large blood vessels and it is characterized by the focal retention and accumulation of fatty materials in the arterial wall in the forms of plaques [5]. The progression of atherosclerosis leads to the ultimate narrowing and hardening of the arterial lumen and the occurrence of a compromised and restricted blood flow [6]. Although the vessel initially attempts to compensate for the luminal occlusion by outward remodeling, this is eventually overcome, and luminal occlusion leads to the clinical condition of angina. The narrowing of the arterial lumen, due to atherosclerotic plaque formation, followed by plaque rupture and atherothrombotic vessel occlusion, leads to life-threatening cardiovascular events, including ischemic stroke and myocardial infarction [6,7]. Many animal models of atherosclerosis have been developed to understand the molecular mechanisms of atherosclerotic plaque formation, development, rupture and related cardiovascular events [8]. In addition, these animal models are also used to assess novel anti-atherosclerotic drugs [9]. Historically, atherosclerosis is induced in these animal models via a cholesterol-/fat-rich atherogenic diet, manipulating cholesterol metabolism-related genes, surgery or introducing diabetes-related risk factors for atherosclerosis.

The most important biological risk factor for atherosclerosis is the (elevated) level of low-density lipoprotein (LDL) in plasma [10]. Mechanistically, LDLs are trapped and retained in the sub-endothelial space of large- and medium-sized arteries by modified proteoglycans with elongated glycosaminoglycan chains, as described in the “response to retention” hypothesis of atherosclerosis [11,12,13,14,15]. Trapped LDL gradually undergo oxidative changes to generate immune reactive oxidized low-density lipoprotein (oxLDL), a clinically relevant and predominant form of modified LDL. This modification stimulates oxidative stress and an inflammatory response in vascular cells, including endothelial cells (ECs), and increases the expression of adhesion molecules, such as intracellular adhesion molecule 1 (ICAM-1), vascular adhesion molecule-1 (VCAM-1), E-selectin, and P-selectin, and the expression of chemotactic proteins, such as monocyte chemoattractant protein 1 (MCP-1) [16,17]. These cytokines promote the recruitment of blood-borne monocytes through the endothelium and into the arterial wall, where the monocytes differentiate into macrophages which engulf oxLDL and become foam cells, the hallmark of atherosclerotic plaques [18,19,20]. The formation of foam cells underpins the development of plaques as they secrete a multitude of mediators of inflammatory processes in the blood vessel wall [21]. The inflammatory response further promotes the incorporation of circulating T-cells and monocytes that further boost arterial inflammation and stimulate vascular smooth muscle cell (VSMC) migration from the tunica media into the sub-endothelial space where they phenotypically modulate, proliferate and secrete extracellular matrix proteins that contribute to plaque expansion [22,23,24]. The most prominent characteristics of advanced human plaques are the following: a large necrotic core, a large number of lipid-laden and activated macrophages, a small amount of VSMCs and derived collagen, defective efferocytosis and inflammation. This is accompanied, and followed, by neovascularization within the plaque, hemorrhages and the development of a thin fibrous cap that weakly separates the underlying thrombogenic plaque from the blood stream. The rupture of this thin fibrous cap leads to intra-luminal thrombosis with sudden blockage of arteries and cessation of blood flow, resulting in life-threatening acute ischemic events, such as myocardial infarction (MI), stroke and sudden death [3,6,23,24,25,26,27,28]. Moreover, calcification in atherosclerotic plaque is another parallel phenomenon occurring to a very variable extent across clinical cohorts [29,30,31]. Vessel wall calcification is characterized by the deposition of calcium ion-containing complexes. Calcification is very significant, especially in the presence of diabetes, and it is a clinical marker of atherosclerosis and leads to thrombus formation and vascular occlusion. A lot of work has been done on calcification to explore its effect on plaque progression and vulnerability [32]. The stiffness associated with heavy calcification reduces the affected vessels’ suitability for stenting, the most common treatment for coronary artery occlusion. Overall, there is a great need for further research to increase our understanding of the occurrence, progression, and mechanisms leading to the clinical manifestations of atherosclerotic lesions and its clinical sequelae and for the identification of efficacious therapeutic interventions. 

In 1908, Ignatowski studied the formation of plaque in the aortic wall of rabbits after being fed a high-cholesterol diet [33]. Later, Nikolai Anitshkov, in St Petersburg (1913), discovered that feeding cholesterol to rabbits would induce atherosclerosis [34] and, since then, therapies targeting hypercholesterolemia have been the main strategy to prevent the development and progression of atherosclerosis. Accordingly, many animal models have been developed to study the development of atherosclerosis including the use of mice, rats, rabbits, birds, swine, dogs, pigs and non-human primates and even zebrafish. Despite many differences; all these animal models share the requirement of high plasma cholesterol for the development of atherosclerosis. The ideal characteristics of animal models include a close resemblance with human anatomy and pathophysiology, easy availability, economical, easy to maintain and handle, and having the potential to be used in medical and pharmaceutical research. A novel animal model to study atherosclerosis is expected to mimic important pathophysiological characteristics of humans but also have similar disease topography [35]. Although atherosclerotic plaques develop in many animals after high cholesterol diet feeding, often the topography of the plaques is not the same as in humans. In addition, an important difference in animal models of atherosclerosis and the human condition is the absence of spontaneous plaque rupture and the sequalae of myocardial infarction, stroke and other cardiovascular complications.

Thus, animal models are very important to study the mechanisms of atherosclerosis as well as exploring new pharmacotherapies directed towards preventing atherosclerosis [36]. Mice represent the largest number of models for atherosclerosis, and their characteristics and use have recently been comprehensively reviewed in detail [8]. However, there are multiple reasons why models other than the mouse would be preferred by researchers, so, to meet this need, in this review we provide a comprehensive presentation of animal models of atherosclerosis (including hamsters, rats, rabbits, zebrafish, pigs and non-human primates), their comparative features and interventional strategies, with the aim of providing a valuable resource for the selection of suitable animal models in translational cardiovascular research and for fostering a deepened understanding of the pathological mechanisms and clinical consequences of atherosclerosis. 

## 2. Animal Models of Atherosclerosis

Animal models provide a foundation to identify mechanisms of the development and progression of atherosclerosis and to explore these mechanisms as potential therapeutic drug targets [9,37,38]. Animal models have allowed researchers to study the development of atherosclerosis at different stages of disease progression. Most animal models of atherosclerosis are based either on feeding Western-type diets or genetic manipulation of genes involved in cholesterol metabolism, like apolipoprotein E (ApoE) or low-density lipoprotein receptor (LDLR). Although various wild-type animal models have been used in the research of atherosclerosis, most contemporary research has been conducted in genetically engineered mice and rabbits, followed by pigs and non-human primates. We have summarized the mechanisms of atherosclerosis development, advantages, limitations, and applications of different animal models in atherosclerosis research in Figure 1. In the following sections, we provide a detailed summary and comparative traits for animal models of atherosclerosis.

When selecting the ideal animal model for the research of atherosclerosis or disease-modifying medications to treat or prevent the development of atherosclerosis, numerous criteria need to be taken into account, due to the complexity and chronic nature of the pathogenesis of atherosclerosis. Strain, gender, diet and length of exposure, circadian rhythm, age, ease of maintenance and comparability to the relevant human situation are a few examples of the characteristics which need to be taken into consideration. There are numerous commonly used atherogenic diets and their composition has been evolving because of new knowledge of the role of diet in plaque development in atherosusceptible animals, which is accelerated by feeding with western-type diets that are heavy in fat and cholesterol. Diets are particularly important for the study of atherosclerosis because animals, in contrast to humans, are relatively resistant to the processes that lead to the development of atherosclerosis in humans. After a lull period in the discovery and development of new agents for the treatment of atherosclerosis and cardiovascular disease there has recently been an upsurge in the identification and exploitation of new targets, based on lipid metabolism and inflammation, and a consideration of suitable animal models, both mouse and non-mouse models, as addressed in this review, which add to the value of atherosclerosis research.

### 2.1. Rat Models of Atherosclerosis

Rat models of atherosclerosis possess certain lipid metabolic characteristics which sit between those of mice and humans [39]. So far, only a small number of studies employing genetic manipulation of ApoE or LDLR genes have been published. In one study, ApoE knockout rats showed early signs of lipid deposition and atherosclerosis after psychological stress induced by occlusal disharmony [40]. ApoE knock-out rats have been generated by transcription activator-like effector nuclease (TALEN)-mediated technology. After being fed a high-cholesterol diet, these rats showed typical pro-atherosclerotic dyslipidemia, but, intriguingly, failed to form atherosclerotic plaques [41]. LDLR knock-out rats, via zinc-finger nuclease (ZFN) technology, showed typical plaque formation after being fed a western-type diet for 34–52 weeks, but did not display lesion formation on regular chow diet even after 64 weeks [42].

ApoE/LDLR single and double knock out rat models, generated using CRISPR/Cas9 technology, showed severe dyslipidemia and atherosclerosis on regular chow diet for 48 weeks, which was further pronounced on feeding with a western-type diet [43]. The three mutant rats did not show differences in their atherosclerotic phenotype. Most importantly, these mutant male rats showed a heavier plaque burden than females, similar to the disease profile in humans [43]. ApoE and LDLR knockout rats, via CRISPR-Cpf1 technology, showed slower progression of atherosclerosis after HFD feeding for 8 weeks [44]. In summary, rat models of atherosclerosis could serve as an ideal animal model to study early stages of human atherosclerosis.

### 2.2. Hamster Models of Atherosclerosis

Hamster models have metabolic similarities and are more comparable to humans because they have cholesteryl ester transfer protein (CETP), a key enzyme in cholesterol and lipid metabolism and, like humans, low cholesterol synthesis in the liver. Recently, LDLR deficient Syrian hamster models have been developed. Heterozygous LDLR deficient hamsters (LDLR^+/−^) showed a marked increase in cholesterol and triglycerides as early as 1 week after feeding a high-fat high-cholesterol diet and these animals developed aortic atherosclerotic lesions in a time dependent manner from 2 to 4 months [45,46]. LDLR deficient hamster models also develop atherosclerosis in a manner which is similar to familial hypercholesterolemic patients [45]. LDLR^+/−^ and LDLR^−/−^ hamsters have similar lipid profiles to human familial hypercholesterolemic patients resulting from LDLR mutations which make them a suitable animal model of atherosclerosis [47].

Moreover, LDLR^+/−^ hamsters have LDLR expression in the liver, and, thus, have the potential to be used for evaluating drug candidates targeting the LDLR pathway. For example, the efficacy of PCSK9 monoclonal antibody, evolocumab in dyslipidemia and atherosclerosis was evaluated in the heterozygous LDLR^+/−^ hamster model. After being fed a high-fat high-cholesterol diet for 12 weeks, evolocumab prevented hyperlipidemia and atherosclerotic plaque formation, suggesting that the LDLR^+/−^ hamster is a suitable animal model to evaluate the therapeutic effects of drug candidates which target PCSK9 [48]. Recently, another hamster model was created by inactivating the Apolipoprotein C-III (ApoC3) gene, using CRISPR/CAS9 technology, to investigate the atherosclerotic effects of the ApoC3 gene [49]. After being fed a high-cholesterol high-fat diet (1.5% cholesterol and 15% fat) for 4 months, ApoC3 knockout (ApoC3^−/−^) hamsters displayed an anti-atherogenic lipid profile and exhibited less atherosclerotic lesions in both abdominal and thoracic arteries, like humans; this data suggested that ApoC3 was a potential therapeutic target for the treatment of atherosclerosis [49]. Thus, hamster animal models are suitable for translational studies of human atherosclerosis and familial hypercholesterolemia and have great potential in the study of atherosclerosis and familial hypercholesterolemia [46].

### 2.3. Rabbit Models of Atherosclerosis

Rabbits have been a frequently used animal model of hyperlipidemia and atherosclerosis because of the similarity of their lipoprotein metabolism to humans, such as the expression of LDL-rich lipoproteins in plasma, CETP and apolipoprotein B mRNA editing (except for the deficiency of hepatic lipase) [50]. Rabbits spontaneously developed atherosclerotic lesions when fed an atherogenic diet for 4 to 8 months. In rabbits, atherosclerotic lesions have been induced by balloon injury in the left iliac artery and feeding atherogenic diet to New Zealand white rabbits. These rabbits developed neointimal thickening with extensive lipid infiltration, high smooth muscle cell content and the presence of macrophage-derived foam cells [51]. Homozygous ApoE knockout rabbits developed mild hyperlipidemia on a chow diet but exhibited greater hyperlipidemia and aortic atherosclerosis on a cholesterol diet (0.3% cholesterol and 3% soybean oil) for 10 weeks [52].

Watanabe heritable hyperlipidemic (WHHL) rabbits are another animal model of spontaneous coronary atherosclerosis and myocardial infarction. Moreover, the phenotype of atherosclerotic plaques in these rabbits is similar to human unstable plaques, such as for example, a thin fibrous cap, advanced lesions with calcification and a necrotic core, thus representing a good model of human plaque rupture [53]. As mentioned above, CETP protein is present in rabbits and plays an important role in lipoprotein metabolism. CETP knockout rabbits, by zinc finger nuclease (ZFN) gene editing, have been generated. Results showed that genetic deficiency of CETP protects against cholesterol diet-induced atherosclerosis in rabbits [54].

Recently, several genetically modified rabbits have been generated. For example, knockout rabbits, using CRISPR-Cas9 along with base-editing [55], and CRISPR/Cpf1 [56]. LDLR knockout rabbits, via CRISPR/Cas9 technology, have been generated and studied. These rabbits spontaneously develop hypercholesterolemia and atherosclerotic lesions in the aorta and coronary artery on a regular chow diet at 12 weeks. These rabbits exhibited increased levels of plasma total cholesterol, LDL-C and triglycerides, accompanied by reduced HDL-C levels, as compared to wild-type rabbits [57]. LDLR/ApoE double knockout rabbit models have been developed, by CRISPR/Cas9 technology, and they showed increased total cholesterol levels up to 10-fold as compared to wild-type rabbits and exhibit hyperlipidemia and atherosclerosis in the aorta as well as the coronary arteries at the age of 12 weeks when fed a regular chow diet [58]. The standard protocol for the generation of knockout rabbits, via CRISPR/Cas9 technology, has been recently published [59] and is further defined by Liu et al. [55,60].

Recently, genetically modified rabbits have been used to study the function of different genes. For example, apolipoprotein (Apo) CIII knockout rabbits were generated, using ZFN technology, to study the atherogenic role of ApoCIII protein. The results indicated that ApoCIII deletion promotes TG-rich lipoprotein catabolism in the liver and its deficiency inhibits hyperlipidemia atherosclerosis in knockout rabbits [61]. Apart from ApoCIII knockout rabbits, human ApoAII knock in rabbits have also been developed. Recently, it has been reported that human ApoAII knock in rabbits, produced using TALEN technology, showed higher HDL-cholesterol levels and lower TG levels accompanied by reduced aortic atherosclerosis on a cholesterol diet [62]. Hypertensive WHHL rabbits were also generated by surgical removal of the left kidney and partial ligation of the right renal artery to study the important role of hypertension in the acceleration of atherosclerosis. The results showed that hypertension enhances advanced atherosclerosis and induces cardiac death in WHHL rabbits within 34 to 56 weeks after surgery, while no death was observed in control rabbits. These results indicated that hypertension not only enhances the development of atherosclerosis but also promotes plaque rupture that leads to cardiac death [63], a phenomenon that is rarely seen in mouse models. Another study also reprised the effect of hypertension on atherosclerosis by showing that hypertension promotes atherosclerosis in large arteries and also affects arteriolosclerosis in small arteries independent of lipid and apolipoprotein levels in plasma [64].

For a detailed review of rabbit models of atherosclerosis, readers are referred to recent reviews published elsewhere [50,65,66]. Overall, rabbit models of atherosclerosis represent useful models to study hyperlipidemia, atherosclerosis and related translational research, as they are phylogenetically closer to humans than mice and rats. However, rabbits are more expensive, need more time to breed and require maintenance of a larger space, as compared with rodents. Furthermore, genetic engineering in rabbits is more technically challenging than in mice.

### 2.4. Other Animal Models of Atherosclerosis

Zebrafish are also an excellent model of hyperlipidemia and atherosclerosis, due to their small size, conserved lipid metabolism, fecundity, rapid development, ex utero fertilization, embryonic transparency and simplicity for experimental manipulation. Different hyperlipidemia and atherosclerotic models of zebrafish have been generated including a high-cholesterol diet induced model, LDLR mutant zebrafish model [67], APOC2 mutant zebrafish model [68] and Liver X receptor (LXR) mutant zebrafish model [69]. These zebrafish models have been recently reviewed by Vedder et al. and Tang et al. [70,71].

Apart from the small animal models, a few large animal models of atherosclerosis, such as pigs and non-human primates, are used, since their cardiovascular system and physiology is like that of humans. In pigs, atherosclerosis is often induced by feeding atherogenic diets or by manipulation of genes involved in lipid metabolism, such as PCSK9, ApoE or LDLR [72]. Ossabaw Pigs with PCSK9 gain-of-function mutation displayed elevated levels of LDL on a regular chow diet compared to wild type (388.7 ± 45.8 mg/dL vs. 29.9 ± 12.6 mg/dL). After feeding a high-cholesterol, high-fat diet for months, their LDL level remarkably increased to 853.6 ± 161 mg/dL and, similar to humans, they showed atherosclerotic plaque formation in the right coronary artery [73]. Similar atherosclerotic plaques were observed in peripheral vessels by another research group [74]. ApoE^−/−^ pigs, generated with CRISPR/Cas9 technology, displayed moderate increases in plasma cholesterol on a regular chow diet but on a high-cholesterol high-fat diet for 6 months they developed severe hypercholesterolemia and atherosclerotic lesions in the aorta and coronary arteries, similar to humans [75]. Similarly, LDLR^−/−^ pigs also displayed significantly higher levels of LDL and lower levels of HDL on a regular chow diet and after 4 months of a high-cholesterol high-fat diet, human-like advanced coronary plaques were developed [76]. There has also been limited use of dogs as models to study atherosclerotic cardiovascular disease and this area has very recently been reviewed [77].

Non-human primates, the biggest animal model in size, represent valuable animal models for human atherosclerosis. They develop hypercholesterolemia and coronary fibro-fatty atherosclerotic lesions after high-cholesterol high-fat diet feeding, which is similar to humans [16,78]. Knockout non-human primates have also been generated, which increase the interest in large animal models with accelerated atherosclerosis [79]. Previously, it has been shown that inhibition of miR-33a/b (repressor of cholesterol transporter ABCA1 and key regulator of HDL biogenesis) by anti-miRNA oligonucleotide in non-human primates increases plasma HDL and decreases VLDL-associated triglycerides over 12 weeks and, thus, inhibits atherosclerosis that is highly relevant to humans [80]. In another study, lipoprotein (a) was knocked down in non-human primate models of atherosclerosis by using lipoprotein (a) siRNA. The results enabled the researchers to unmask the role of lipoprotein (a) in atherosclerosis development and progression in primates [81]. Recently, the efficacy and safety assessment of tumor necrosis factor receptor-associated factor 6 (TRAF6)-targeted nanoimmunotherapy was assessed in ApoE^−/−^ mice and non-human primates. The aim of this nanoimmunotherapy was to block the interaction between CD40 and TRAF6. The in vivo results of positron-emission tomography imaging showed that 1-week nanoimmunotherapy treatment significantly reduced plaque inflammation, and suggested that this therapy had the translational potential for the treatment of atherosclerosis [82]. Overall, large animal models of atherosclerosis, such as non-human primates and pigs, represent valuable and clinically relevant models for testing therapeutic interventions and insightful translational studies of atherosclerosis.

## 3. Interventional Strategies for Preclinical Therapeutic Studies of Atherosclerosis

### 3.1. Administration of Drugs or Compounds to Attenuate Atherosclerosis

There are numerous examples of the successful administration of drugs to animals which attenuate, to various degrees, lipid deposition, atherosclerosis and changes in the vasculature of animals with experimental models of vascular disease [9,38,83,84]. The appropriate route of drug administration is crucial in determining the pharmacodynamics, pharmacokinetics and toxicity of the drugs [85]. Intraperitoneal (*i.p.*), intravenous (*i.v.*) and oral routes (*p.o.*) are commonly used routes of drug administration in animal models of atherosclerosis and each route has its own benefits and limitations according to the specific objectives of the study. The intraperitoneal injection route, where a drug is injected into the peritoneal cavity, is most commonly practiced in animal models, as it is easy, quick and a suitable route with low impact of stress on the animal. The IP route provides a faster rate of drug absorption (compared to oral administration) and is also well-suited for drug suspensions and solutions [86]. However, it also comes with disadvantages, like high first-pass metabolism, the need for sterility and limited space for multiple injections. The intravenous route results in the highest bioavailability of a drug as compared to all other routes with no obvious limitation in absorption and rapid onset of action. However, the intravenous route is often challenging for rodent studies, as it needs advanced skills to practice (due to the small size of the mouse tail vein), is less suitable for chronic/repetitive procedures and is also impractical because most of the drugs are hydrophobic in nature and are difficult to dissolve in aqueous solutions. The oral route is easy, convenient, and suitable for repeated administration. However, drugs using this route have relatively low absorption and poor bioavailability and can also damage the esophagus, which leads to reduced food intake and even mouse death [86]. In addition, some of the drugs/compounds are not orally bioavailable, and, thus, not suitable for oral gavage. Supplementation of drugs in the diet or drinking water is another option for oral administration of drugs. However, the dose of drug administered to individual mice is hard to calculate, which may lead to unexpected experimental variations.

### 3.2. Nucleic Acid-Based Drug Delivery

Nucleic acid-based technologies, including short hairpin (sh) RNA, small interfering (si) RNA, specific antisense oligoneucleotides (ASO), and locked nucleic acid (LNA), represent emerging therapeutic strategies to treat atherosclerosis. The aim of these strategies is sequence-specific gene silencing of certain atherogenic transcripts, resulting in an ultimate decrease of atherogenic lipoproteins [87]. For example, morpholinos are synthetic ASO analogs used to silence atherogenic genes in vivo. They bind to complementary RNA sequences and promote targeted gene degradation (ribonuclease H1 dependent class) or block their translation or pre-mRNA splicing (steric-blocker class) [88]. They are alternatives to siRNA, with high specificity and low antisense effects. Recently, morpholinos have been effectively used to silence different anti-atherogenic or atherogenic genes and to explore their role in disease development. For example, sequence specific morpholinos were used to block YAP/TAZ translation in vivo to investigate its atherogenic functions [89]. Similarly, morpholinos were used to knockdown Zfp36l1b in zebrafish, to explore its protective role in angiogenesis [90]. Furthermore, sphingolipid inhibitor d-PDMP (d-threo-1-phenyl-2-decanoylamino-3-morpholino-1-propanol, a glycosphingolipid inhibitor), has been reported to reduce lipid uptake and vascular inflammation and to protect against atherosclerosis in ApoE^−/−^ mice [91]. In another study, antisense oligonucleotides targeting receptor-interacting serine/threonine-protein kinase 1 (RIPK1) in ApoE^−/−^ mice identified the inflammatory role of RIPK1 in atherosclerosis and suggested RIPK1 as a potential future therapeutic target to limit atherosclerotic inflammation [92]. For the nucleic acid drugs to reach the target site (atherosclerotic plaques, in particular) to be functional, biochemical modification is necessary, such as the GalNAC modification to target the liver, and some liposome nanoparticles (LNP) to reach the plaque resident cells.

### 3.3. CRISPR-Cas9 Technology

Genome editing, via CRISPR-Cas9 technology, is an emerging effective approach to edit mutated genes associated with disease development. For example, knockout of the PCSK9 gene via CRISPR-Cas9 resulted in 30% reduction of LDL cholesterol in mice, without off-target mutagenesis [93]. Another study has reported that base editing targeting of human and mouse PCSK9 via CRISPR-Cas9 resulted in significant reduction in plasma total cholesterol level [94]. In addition, base editing of ANGPTL3, via CRISPR-Cas9, in LDLR^−/−^ mice reduced plasma TG and TC levels by 56% and 51%, respectively [95]. Furthermore, in vivo AAV-CRISPR/Cas9-mediated LDLR gene editing partially rescued LDLR expression, reduced TC, total TG, and LDL-cholesterol level and effectively ameliorated atherosclerosis phenotypes in mouse models of FH [96]. Overall, genome editing technology, via the CRISPR-Cas9 system, is yielding promising results in experimental models of atherosclerosis. One additional notable feature of this technology is that researchers can achieve global deletion or conditional deletion of genes of interest or transgenes directly in atherosusceptible mouse strains without backcrossing with ApoE^−/−-^ or LDLR^−/−^ mice. Thus, CRISPR-Cas9 technology stands out as a time- and cost-efficient way of studying atherosclerosis.

## 4. Conclusions and Perspectives

Atherosclerosis and its clinical consequences are the leading cause of morbidity and mortality in developed countries. Currently, the majority of contemporary experimental and pre-clinical studies are conducted in murine models of atherosclerosis. However, there is a big difference between mouse and human atherosclerosis and clinical translatability varies considerably. So far, no perfect animal model can fully reproduce, with high fidelity, the properties of human atherosclerosis and its multifactorial characteristics. On the basis that atherosclerotic plaques are the major clinically relevant pathophysiological feature of human atherosclerosis, then the ideal animal model of atherosclerosis would develop lesions which were comparable to human lesions. Most of the available animal models of atherosclerosis are useful to study lesion development and to explore the effectiveness of therapeutic agents. However, differences in anatomy, gene expression and lipid metabolism complicate translation of experimental results obtained in these animal models to humans. Moreover, in most of the animal models, hypercholesterolemia is the major triggering factor of lesion development, but it is multifactorial in humans. Mouse models also display higher inflammation within the vasculature, as compared to humans, and, thus, targeting inflammatory pathways has a more pronounced anti-atherogenic benefit in mice, which has proven not to be translatable to humans. Another major limitation of animal models of atherosclerosis is the formation of atherosclerotic plaques in the aorta and proximal large vessels and very rarely plaque formation in coronary vessels, a feature that is not the case in humans. To mimic the human atherosclerosis pathophysiology in animal models, and for translational potential, a more suitable animal model with atherosclerotic plaques in the coronary artery is required. Another big shortcoming of animal models is the rarity of plaque rupture, thrombosis, end-stage ischemic lesions, myocardial infarction and stroke, which are the most commonly occurring events in humans.

Moreover, most of the atherosclerosis studies have been conducted on male animals, which neglects the role of gender-specific effects in atherosclerosis. So, it is recommended to include both sexes in the study design as recommended by the “NIH Policy on Sex as a Biological Variable” to increase the confidence, rigor and reproducibility of atherosclerosis research. Overall, in terms of pathophysiology, all available animal models of atherosclerosis show some, but not all, the characteristics of human plaques. It is urgent, but not easy, to develop new suitable models that mimic the complicated human plaque, with features of neovascularization, calcification, intraplaque hemorrhage, and thrombosis, Such models could be used to develop more potential therapeutic agents for plaque stabilization and to enhance robustness and scientific rigor. Likewise, animal models of atherosclerotic plaque regression need to be developed that can reverse the pre-existing plaques, as regression of atherosclerosis is clinically and therapeutically desirable. In addition, in patients with atherosclerosis, many co-morbidities may co-exist, and, therefore, complex animal models of atherosclerosis, and its co-morbidities (such as diabetes and hypertension), are required to unravel the mechanisms of atherosclerosis and to achieve the aim of developing safe and effective therapeutic agents for this major human disease.

## Figures and Tables

**Figure 1 ijms-23-12964-f001:**
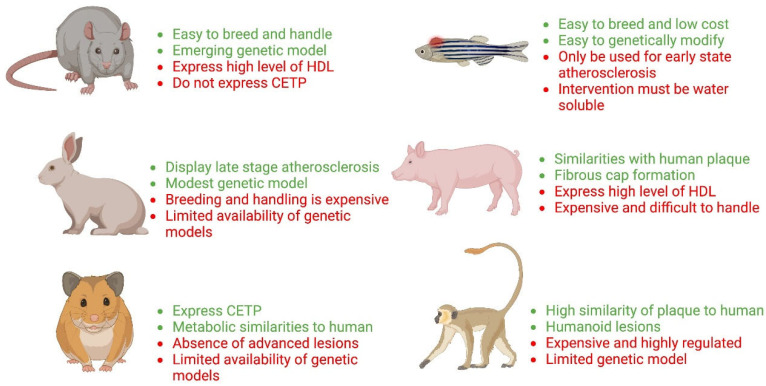
Main characteristics of non-mouse animal models of atherosclerosis. Green colour indicates advantages while red colour indicates disadvantages of these animal models.

## Data Availability

Not applicable.

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
