# Peer review of "Non-Mouse Models of Atherosclerosis: Approaches to Exploring the Translational Potential of New Therapies"

_ijms, 2022, doi:10.3390/ijms232112964_

Round 1
Reviewer 1 Report
Kamato et al. provided an overview of the advantages and shortcomings of various non-mouse animal models of atherosclerotic plaque formation, plaque rupture as well as commonly used interventional strategies. This review provided a comprehensive presentation of animal models of atherosclerosis (including hamsters, rats, rabbits, zebrafish, pigs and non-human primates), their comparative features and interventional strategies, with the aim of providing a valuable resource for the selection of suitable animal models in translational cardiovascular research and for fostering a deepened understanding of the pathological mechanisms and clinical consequences of atherosclerosis. The combinatorial selection of suitable animal models will readily facilitate reproducible and rigorous translational research in discovering and validating novel anti-atherosclerotic drugs. This study is interesting.
1. Mice are the common animal models of atherosclerosis. It is better that other models should be compared with mice, and relevant features should be discussed among different models.
2. It is better if more Figures and/or tables are presented to show the detailed features for non-mouse animal models, or the detailed interventional strategies or involved molecular mechanisms.
3. The Figure 1 is unclear and it should be improved.
4. The second part of “Animal models of atherosclerosis” should be further enriched, which is the main contents in the study.
Author Response
Thank you for positive comments. |
This matter is addressed in the Figure (which is now new). |
As below we have improved the Figure 1 – this is a medium length paper and we hope the reviewer will be happy if we leave it at one good figure. |
A new improved Fig 1 has been produced and is now included in the revised manuscript. |
This new text has been added: When selecting the ideal animal model for the research of atherosclerosis or disease-modifying medications to treat or prevent the development of atherosclerosis, numerous criteria need to be taken into account due to the complexity and chronic nature of the pathogenesis of atherosclerosis. Strain, gender, diet and length of exposure, circadian rhythm, age, ease of maintenance and comparability to the relevant human situation are a few examples of the characteristics which need to be taken into consideration. There are numerous commonly used atherogenic diets and their composition has been evolving because of new knowledge of the role of diet in plaque development in atherosusceptible animals which is accelerated by feeding with western-type diets that are heavy in fat and cholesterol. Diets are particularly important for the study of atherosclerosis because animals, in contrast to humans, are relatively resistant to the processes that lead to the development of atherosclerosis humans. After a lull period in the discovery and development of new agents for the treatment of atherosclerosis and cardiovascular disease there has recently been an upsurge in the identification and exploitation of new targets based on lipid metabolism and inflammation and a consideration of suitable animal models, both mouse and non-mouse models as address in this review will add to the value of atherosclerosis research.
|
Reviewer 2 Report
The reviewed manuscript is a synthetic and useful review of animal models of atherosclerosis, paying attention to the important and useful features of the models of individual species and their imperfections in relation to the human form of this disease. Authors also included those models that emerged from the development of new gene editing techniques. The information contained in the manuscript may be very useful for researchers considering the choice of the model of their research and become a starting point for searching for more detailed literature on selected models. The Authors emphasized that none of the described models is ideal for humans and allows for research on only a narrow range of changes caused by this disease. The Authors also drew attention to the important aspect of the methods of drug administration in this disease - what disadvantages in this respect are in different animal models and also described modern technologies for the treatment of atherosclerosis. An important issue raised by the Authors was also to draw attention to the necessity to undertake research on females - since sex differences are important for its course.
Minor comments:
Line 100 – there are animals mentioned called “dons” – is it correct? Perhaps “dogs”?
Line 253 – the word “have” is unnecessary
Line 344 – short explanation what YAP/TAZ is would be useful.
Line 346 – abbreviation “Zfp36l1b“ should be explained.
Author Response
Thank you for the positive comments. |
|
Found and corrected – thank you. |
Line 271 in revised manuscript – the word “have as been removed – thank you. |
We have added that “YAP/TAZ pathway is associated with angioigenesis such that Hippo pathway-YAP/TAZ is now demonstrated to regulate endothelial cell proliferation, migration and survival; subsequently regulating vascular sprouting, vascular barrier formation, and vascular remodeling”. |
We have added the full names: zinc finger protein 36, C3H type-like 1 |
Reviewer 3 Report
In this review Kamato aims to provide an overview of the advantages of various non-mouse animal models of atherosclerotic process.
However, recent work on this topic has already been published (Circ Res. 2022 Jun 10;130(12):1869-1887. doi: 10.1161/CIRCRESAHA.122.320263), including “Non-mouse models of atherosclerosis”
To this end, in order to improve the novelty of paper, authors could provide an additional paragraph elucidating the methods and epigenetic modifications to quantify atherosclerotic lesions, including oxidative stress and inflammatory response occurring in endothelial cells, in experimental atherosclerosis models.
Figure 1 is very low resolution
Additional figures must be provided to explain the experimental Non-mouse models of atherosclerosis and improve the paper.
Author Response
Thank you. |
|
Thank you for recognizing the standing of our review. . |
There are multiple figures and tables in the companion paper in Pharmacological Research (https://doi.org/10.1016/j.tips.2022.06.009) which was published recently and is referenced in our current paper. The only reason that there are two papers is because the material was too large to be included in one paper. |
Reviewer 4 Report
The authors present a review article which aims to identify and evaluate potential novel cardiovascular drugs, suitable animal models that reproduce human atherosclerosis with high degree of fidelity are required as essential pre-clinical research tools.
Comments:
The Non-mouse models of atherosclerosis is important.
Maybe a summarize Table or Figure of “interventional strategies for preclinical therapeutic studies of atherosclerosis” should be provided in this review article.
Author Response
Thank you. |
|
Thank you for recognizing the standing of our review. . |
As below we have improved the Figure 1 – this is a medium length paper and we hope the reviewer will be happy if we leave it at one good figure. |
Round 2
Reviewer 1 Report
The manuscript has been carefully checked and revised. The improved manuscript can be acceptable for publication in the journal. Thanks.
Reviewer 4 Report
No further comment.